# Performance Profile among Age Categories in Young Cyclists

**DOI:** 10.3390/biology10111196

**Published:** 2021-11-17

**Authors:** Cristian Marín-Pagán, Stéphane Dufour, Tomás T. Freitas, Pedro E. Alcaraz

**Affiliations:** 1UCAM Research Center for High Performance Sport, Catholic University of Murcia (UCAM), 30107 Murcia, Spain; tfreitas@ucam.edu (T.T.F.); palcaraz@ucam.edu (P.E.A.); 2Faculty of Medicine, Translational Medicine Federation (FMTS), University of Strasbourg, EA 3072 Strasbourg, France; sdufour@unistra.fr; 3Faculty of Sport Sciences, University of Strasbourg, 67000 Strasbourg, France; 4NAR-Nucleus of High Performance in Sport, São Paulo 04753-060, Brazil; 5Faculty of Sport Sciences, Catholic University of Murcia (UCAM), 30107 Murcia, Spain

**Keywords:** cycling, endurance, oxygen uptake, FTP, threshold, power

## Abstract

**Simple Summary:**

Overall, adolescence brings upon many bodily changes that modify physical capacities. To better understand these physiological changes and the characteristics of each stage of adolescent development in youth cycling, it is necessary to describe and compare cyclists that pertain to lower categories. Parameters such as maximum oxygen uptake, fat oxidation capacity, functional power threshold, and ventilatory thresholds are decisive predictors of performance in future stages. The aim of this study was to evaluate and compare the physiological profile of different road cyclist age categories (Youth, Junior, and Under-23) to obtain the performance requirements. The results suggest major differences, with the Youth group showing clear changes in all metabolic zones except in fat oxidation. The Youth group physiological profile is clearly different from the other age categories. The present results suggest that the Juniors’ qualities are closer to adult performance, however, little is known about sports performance indicators in adolescent cyclists.

**Abstract:**

Endurance profile assessment is of major interest to evaluate the cyclist’s performance potential. In this regard, maximal oxygen uptake and functional threshold power are useful functional parameters to determine metabolic training zones (ventilatory threshold). The aim of this study was to evaluate and compare the physiological profile of different road cyclist age categories (Youth, Junior, and Under-23) to obtain the performance requirements. Sixty-one competitive road cyclists (15–22 years) performed a maximal incremental test on a bike in order to determine functional parameters (maximal fat oxidation zone, ventilatory thresholds, maximal oxygen uptake, and functional threshold power) and metabolic training zones. The results suggest major differences, with the Youth group showing clear changes in all metabolic zones except in fat oxidation. The main differences between Under-23 vs. Junior groups were observed in maximal relative power output (Under-23: 6.70 W·Kg^−1^; Junior: 6.17 W·Kg^−1^) and relative functional threshold power (Under-23: 4.91 W·Kg^−1^; Junior: 4.48 W·Kg^−1^). The Youth group physiological profile is clearly different to the other age categories. Some parameters normalized to body weight (maximal oxygen consumption, load and functional threshold power) could be interesting to predict a sporting career during the Junior and Under-23 stages.

## 1. Introduction

Cycling is considered one of the most stressful and physically demanding sports, with the road stage races being its most popular modality. In professional road cyclists, values for maximal oxygen uptake (VO_2_max) higher than 70–80 mL·Kg^−1^·min^−1^ have repeatedly been observed [1,2]. Although very high, such VO_2_max values appear more as a prerequisite to achieve professional level rather than good performance predictor [3]. As such, maximal power output during an incremental test might be a better predictor than VO_2_max for short efforts in flat stages, with elite cyclists achieving values between 400 and 500 W (6.0–7.5 W·Kg^−1^), finding slight differences depending on the test characteristics [4,5]. Additionally, lactate threshold position seems to be more predictive than VO_2_max for endurance cycle performance, especially in professional climber cyclists [5] where lactate thresholds (LT2) at ~90% of VO_2_max have been found.

In recent years, the evaluation of cycling performance and the monitoring of cycling training load through the so-called functional threshold power (FTP) has also been of increasing scientific interest [6,7,8]. FTP consists of the maximum power output developed during a 1 h trial, and can be evaluated by contemplating the power developed during 20 min with the application of a correction factor of 0.95 [9]. FTP values are estimated to be around 5.0–6.0 W·Kg^−1^ and 3–5 W·Kg^−1^ for professional and trained amateur cyclists, respectively [9]. The FTP has become a supplementary parameter to the assessment of performance profile due to its applicability to the field in a non-invasive way and without the need for sophisticated equipment.

Achieving professional and world-class level in road cycling is a long-term process taking several years of regular, high volume, and high intensity training, most of the time from Youth, through to the Junior and Under-23 (U-23) categories. Therefore, important differences in physiological profile exist between competition levels and age categories in cycling [10]. For example, professional road cyclists complete approximately 30,000 to 35,000 km per season [4] while amateur competitive cyclists complete around 13,500 km [11]. Important changes in total volume progression have also been observed when comparing consecutive seasons from the Junior stages to World class level [12]. Similarly, higher VO_2_max values have been reported in elite cyclists (~74 mL·Kg^−1^·min^−1^) when compared to amateurs (~65 mL·Kg^−1^·min^−1^) [13,14]. Another critical factor that has been suggested to differentiate cyclists of superior performance levels [5,13,15] is the ability to develop power, both as peak values in incremental tests or during critical power [16] tests as a FTP [9]. Due to mentioned discrepancies between cyclists, De Pauw et al. [13] proposed a five level cycling classification according to physiological demands and training loads. Nevertheless, it is worth noting that amateur cyclists are progressively showing higher performance levels, to the extent that similarities with professionals can be found, particularly in cycling economy and efficiency [17].

Regarding age categories, previous studies have reported differences in anthropometric parameters, with athletes displaying greater left–right leg length asymmetries as they progress through to older categories [18]. This unbalance could be related to an increased training duration in more experienced cyclists to meet the demands of the competition. The characteristics of Youth and Junior races are different and very stressful for the metabolic system [19] and some countries have limited the maximal number of competitions per season during the Youth categories. As the distances are usually shorter in these categories, the average race intensity is higher. For this reason, due to the progression in training volume [12] and competition characteristics [19] the recovery time necessary after an endurance exercise increases with age [20].

On a related topic, during prolonged training and competitive efforts (>4 h), an increased fatty acid contribution to total energy turnover is observed and fat oxidation capacity could be considered as a desirable adaptation for road cycling performance [21,22]. Accordingly, assessing and training to improve this capacity are important aspects of the training process in cycling [22]. Different authors have reported that the maximal fat oxidation zone (Fatmax) is achieved at approximately 45–60% of VO_2_max [23,24] and that this concept is closely related to cycling economy and efficiency. The issue with the former variables is that they are somehow easy to improve in amateurs, but very difficult when it comes to professional cyclists. Thus, in highly trained cyclists, it is common to observe that endurance training is not sufficient to improve cycling economy and efficiency, which makes it necessary to rely to alternative training strategies such as resistance training to achieve this objective [25,26], with heavy strength training being recommended to achieve improvements in aerobic performance [26,27].

From the above-mentioned, it appears that endurance performance parameters are likely different among the Youth categories, but the extent of these differences remain presently unclear. Dissimilarities in some cardiorespiratory and metabolic parameters between professionals and amateur cyclists [13,14] as well as between age categories [14] can be found in the literature. However, the differences in physiological and performance profile of three different age categories (Youth, Junior, and U-23) in the pre-season phase as well as the value of FTP in these three age categories have never been documented to date. Such knowledge might have important practical applications not only for better performance evaluation (i.e., talent identification) but also to optimize training prescription and training load management in young cyclists. Currently, there are no studies that compare the physiological profile in these categories, in order to establish differences or similarities that could help determine future performance. It is also unknown which are, in these lower categories, the most important physiological characteristics to be studied and controlled. Therefore, the aim of this study was to assess the physiological parameters and performance profile that can influence cycling performance and to compare them amongst three different age categories, from Youth to U-23.

## 2. Materials and Methods

### 2.1. Participants

Sixty-one young male amateur cyclists from three distinct age categories, but with similar competitive level, participated in the study (Table 1): Youth (15–16 years); Junior (17–18 years), and U-23 (19–22 years), according to the classification of the *Union Cycliste Internacionale* (UCI). All cyclists were members of an official team, did not present any injury in the three months before the investigation and performed regular training of more than 6 h per week. Participants had at least three years of cycling training experience, were enrolled in the cycling team since “school” categories, and had previous experience in laboratory testing. Prior to study enrollment, all cyclists or the parents (of those under 18 years old) signed the consent to participate in the study (approved by the University Ethical Committee; CE022105) and obtained medical approval to participate in this study. Just before testing, weight and height were measured using a SECA 780 device (Seca, Hamburg, Germany). All tests were completed between 10 h and 14 h 2 h after breakfast intake (bread, milk or yogurt and juice). Participants did not train in the 24 h prior to testing to avoid fatigue and the test was separated from high intensity training or pre-season competitions by at least 72 h.

### 2.2. Assessments

All tests were carried out in the laboratory during pre-season (December–February). For the cardiorespiratory evaluation, a metabolic cart (Cortex Metalyzer, Leipzig, Germany) and the Cyclus2 ergometer (Cyclus, Leipzig, Germany) were used. The cyclists utilized their own bikes in all assessments. The protocol used consisted of a combined test with an initial step phase followed by final ramp. The test started at 35 W with increments of 35 W every 2 min. Then, when the respiratory exchange ratio (RER) was ≥1.05, the final ramp of 35 W per minute (~1 W each 0.583 s) was initiated. This combined protocol was applied to determine the ventilatory thresholds (VT1 = aerobic; VT2 = anaerobic) during steady states (step phase) and continued until exhaustion to assess VO_2_max and maximal load (Pmax, final ramp) [28,29,30]. The recommended pedaling cadence was 85 to 95 rpm and the test was stopped when the participants were unable to sustain a cadence greater than 60 rpm, with permanent chainset (52–53/12 teeth). To determine blood lactate concentration, blood samples were collected from the finger at 1.5 min after exhaustion. The first blood drop was dismissed and the second was analyzed with a Lactate Pro2 (Arkray, Tokyo, Japan).

Ventilatory thresholds (VT1 and VT2) were calculated with the ventilatory equivalent method described by Wasserman [31] and using the data averaged every 20 s. The VO_2_max was assumed as the maximum value of the last four data of 20 s averages. To guarantee that the VO_2_max was achieved, at least three of the following criteria had to be obtained: (I) plateau in the final VO_2_ values (increase ≤2.0 mL·kg^−1^·min^−1^ in the two last loads); (II) maximal theoretical HR (220–age) × 0.95) for a cycling test suggested by Millet et al. [32]; (III) RER ≥1.15; and (IV) a lactate value ≥8.0 mmol·l^−1^ [33]. Pmax was calculated as the maximal power achieved during the final ramp in the incremental test. Maximal oxygen uptake and load were expressed in absolute units or normalized to body weight (VO_2_R and Load/BW, respectively). To determine the percentage of VO_2_max at which Fatmax was achieved, the values of VO_2_ corresponding to maximal fat oxidation (MFO) and normalized to VO_2_max were selected.

Functional threshold power (FTP) was estimated using the equation described by Denham [34] using the maximal power output during VO_2_max test.

### 2.3. Statistical Analysis

All descriptive statistics were presented as mean ± standard deviation (SD) and the statistical analysis was performed using the Statistical Package for Social Sciences (SPSS 27.0, IBM, Chicago, IL, USA). A Shapiro–Wilk test was performed to assess the normality of the variables. The between-group differences were investigated using independent *t*-tests and the statistical significance was set for a *p* < 0.05. The U-23 group was established as the “reference group” for the comparative analysis, given that it was the highest competitive level. Effect sizes (ES) were calculated utilizing Cohen’s equations [35]. Threshold values for ES statistics were: >0.2 small, >0.6 moderate, and >1.2 large, >2.0, very large; and >4.0, nearly perfect [36].

## 3. Results

*Ventilatory Threshold 1*. Significant differences were obtained between the Youth and Junior groups for VO_2_ (*p* = 0.001; ES = 0.98), Load (*p* < 0.001; ES = 1.21) and Load/BW (*p* = 0.037; ES = 0.62). For Youth vs. U-23 group, significant differences were obtained in HR (*p* = 0.018; ES = 0.80), %VO_2_max (*p* = 0.005; ES = 0.97), Load (*p* < 0.001; ES = 1.32), and Load/BW (*p* = 0.002; ES = 1.06). For Junior vs. U-23, differences were only found in HR (*p* = 0.027; ES = 0.77). In this metabolic zone, the U-23 group showed the lowest percentage with respect to VO_2_max, and the Youth group displayed the best results (Table 2).

*Ventilatory Threshold 2*. For the Youth vs. Junior group comparison, significant differences were found in VO_2_ (*p* < 0.001; ES = 1.47), VO_2_R (*p* = 0.020; ES = 0.70), Load (*p* < 0.001; ES = 1.51) and Load/BW (*p* = 0.011; ES = 0.77). There were also significant differences between the Youth and U-23 groups in VO_2_ (*p* = 0.001; ES = 1.22), VO_2_R (*p* = 0.007; ES = 0.92), Load (*p* < 0.001; ES = 1.80), and Load/BW (*p* < 0.001; ES = 1.40). As for VT1, significant differences were found only in HR (*p* = 0.035; ES = 0.86) when comparing the Junior and U-23 groups (Table 2).

*Maximal Zone*. Significant differences were obtained between the Youth and Junior groups (Table 3 and Figure 1) for VO_2_ (*p* < 0.001; ES = 1.42), VO_2_R (*p* = 0.003; ES = 0.92), Load (*p* < 0.001; ES = 1.51), Load/BW (*p* = 0.003; ES = 0.89), time to exhaustion (*p* < 0.001; ES = 1.67), and blood lactate concentration (*p* = 0.007; ES = 0.86). Similar results were obtained for Youth vs. U-23 for VO_2_ (*p* < 0.001; ES = 1.18), VO_2_R (*p* = 0.002; ES = 1.08), Load (*p* < 0.001; ES = 1.95), Load/BW (*p* < 0.001; ES = 1.88), and time to exhaustion (*p* < 0.001; ES = 1.70) but not for blood lactate (*p* = 0.059; ES = 0.77), in which only a trend toward statistical significance was found. Finally, for Junior vs. U-23, significant differences were found for HR (*p* = 0.013; ES = 1.24) and Load/BW (*p* = 0.015; ES = 0.86).

*Fatmax zone*. Significant between-group differences were found in this metabolic zone (Table 4) only in VO_2_ (*p* = 0.007; ES = 0.86) and Load (*p* = 0.011; ES = 1.00) for Youth vs. U-23.

*Estimated functional threshold power*. For the estimated FTP, significant differences were found (Table 5) between the Youth and the other two groups (Junior and U-23, *p* < 0.001; ES = 1.50 and 2.66, respectively). Additionally, for Junior vs. U-23, a significant difference was obtained in FTP/BW (*p* = 0.014; ES = 0.85).

The FTP normalized to BW is a key factor in cycling, which is related to other performance parameters such as VO_2_max and power output in VO_2_max. Finally, in FTP/BW, a linear increase was found in relation to the age category (R^2^ = 0.995; Figure 2). Additionally, the percentage of FTP with respect to the maximal power output (Pmax) showed a significant difference with the Youth group (Junior and U-23, *p* < 0.001; ES = 1.41 and 1.69, respectively), finding higher values in both groups (4% in Junior and 5% in U-23), but no differences were observed between the Junior and U-23 groups.

## 4. Discussion

The aim of the present study was to assess and compare the physiological profile of different age categories. Despite cardiorespiratory testing being the most frequent procedure to assess performance in cyclists regardless of the level of competition [37], this study is the first to directly compare the Youth, Junior and U-23 categories. The main findings indicated that, during a maximal test, Junior, and U-23 group obtained values of VO_2_max were slightly lower than those reported in elite and professional cyclists [1,2,37], but significantly greater than the Youth group. Similar differences were found for all performance variables analyzed in the maximal effort zone with important results in the ES analysis. These results were somehow expected due to the changes in physiological parameters with age and maturation, but could also be influenced by the athlete’s training background.

Maximal values of VO_2_max and Pmax showed important differences with the Youth (VO_2_max = 8.2–10.2%; Pmax = 13.8–24.8% for Junior and U-23, respectively) and only for Pmax/BW were found differences between the Junior and U-23 (8.6% greater in the U-23 group). The VO_2_max data obtained by the Junior and U-23 groups were similar to the values reported in professional cyclists [1,2]. However, in recent years, relative power production has proven to be a more sensitive indicator, since in professional categories, this parameter allows for better differentiating performance levels when compared to the VO_2_max [4,5]. Along these lines, the present results indicated that the U-23 group outperformed the Youth (+19.0%) and the Junior (+8.6%) categories. For this reason, maximum Load/BW could be an important indicator of the competitive level in the U-23 category.

The FTP was at ~70% of Pmax with differences with the Youth group (4% in Junior and 5% in U-23). Interestingly, great differences between groups were found for Load/BW and FTP/BW, with a linear increase from Youth (19% and 14%, respectively, for Junior) to the U-23 category (25% in both for U-23). These parameters (Load/BW and FTP/BW) have been proposed to be very important for cycling performance [6,7,8,38,39]. Due to the duration of the stages, time under muscle tension is usually large, potentially explaining why power output normalized to BW is crucial for road cyclists. From an applied perspective, FTP/BW could be used at the beginning of the season to determine performance levels and compare them with the reference values of each category. Of note, the FTP assessments were calculated indirectly in our study using the equation proposed by Denham et al. [34], where there were similar values were obtained with direct assessments by the U-23 group in comparison with previously reported values for professional cyclists [3], which supports the notion that the athletes in the studied sample were of high competitive level. It is likely that the FTP could be the most differentiating variable, showing a large and very large ES favoring the U-23 vs. the Junior and Youth, respectively.

Regarding VT1, similarities were found between the Junior and U-23 cyclists and both groups presented greater values than the Youth (especially for Load variables with ~13% in Junior and ~16% in U-23). According to Lucia et al. [1], it is important to achieve a good “cruising speed” in this metabolic zone, because it is the predominant intensity during flat stages. The differences reported herein could be conceivably explained by the competition characteristics in the Youth categories, which are usually comprised of shorter stages. For this reason, the lower intensity profile could be optimized in Junior and U-23 and, hence, closer to the values found in professional and elite cyclists [1,2].

In the work load at VT2, similar differences to those obtained for VT1 between age categories were found, but the values reported for U-23 were clearly lower than for professional cyclists [3]. Although there were no differences between age categories in the VT2 position with respect to VO_2_max, the workload developed in this metabolic zone is crucial to determine the aerobic capacity, which characterizes the professional cyclists [3,39]. Therefore, from an applied perspective, developing the aerobic capacity in younger categories should be an important objective.

Finally, when analyzing the Fatmax zone, similar results were displayed by all age groups with only small differences found between the U-23 and Youth groups for VO_2_ and Load. Notably, these differences were not found for the same parameters when the values were normalized to BW (VO_2_R and Load/BW), which is in line with previous findings [23,24] and had not been reported for either differences in economy and efficiency when amateur and professional cyclist were compared [17]. Based on the present results, coaches should be aware that Fatmax does not seem to be a key factor discriminating the performance level in cyclists, although it could be important in long-term modalities as demonstrated in Ironman triathletes [22].

The workload was found to be the main difference in both thresholds (VT1 and VT2) with respect to the Youth group, as displayed by the large ES. Moreover, workload seems to be the main performance determinant in both maximum and submaximal zones with respect to the Youth group. These results are supported by the large ES obtained. However, when comparing the Junior and U-23, these differences were less pronounced and are only manifested in FTP/BW and maximal Load/BW with moderate ES. Practitioners should be aware of these findings when managing workload and monitoring training through power output zones.

The main limitation of the study was the reduced number of participants. Moreover, the fact that no previous investigations have compared, among the same age categories, the physiological parameters analyzed herein limited the discussion of the findings. Future studies with longitudinal research designs comparing the evolution of the same cyclists during their career would be interesting to conduct.

## 5. Conclusions

The main findings in this study showed enough differences with the Youth group and minor changes in the Junior vs. U-23 group. These results suggest that the main ladder is from the Youth to Junior age category. Due to the minor differences obtained between the Junior and U-23 categories, it could be intuited that the physiological profile in the Junior stage could be predictors of performance in absolute categories. Additionally, FTP/BW showed clear differences between each age category and testing it could be a good method to determine the cycling potential for cyclists.

## Figures and Tables

**Figure 1 biology-10-01196-f001:**
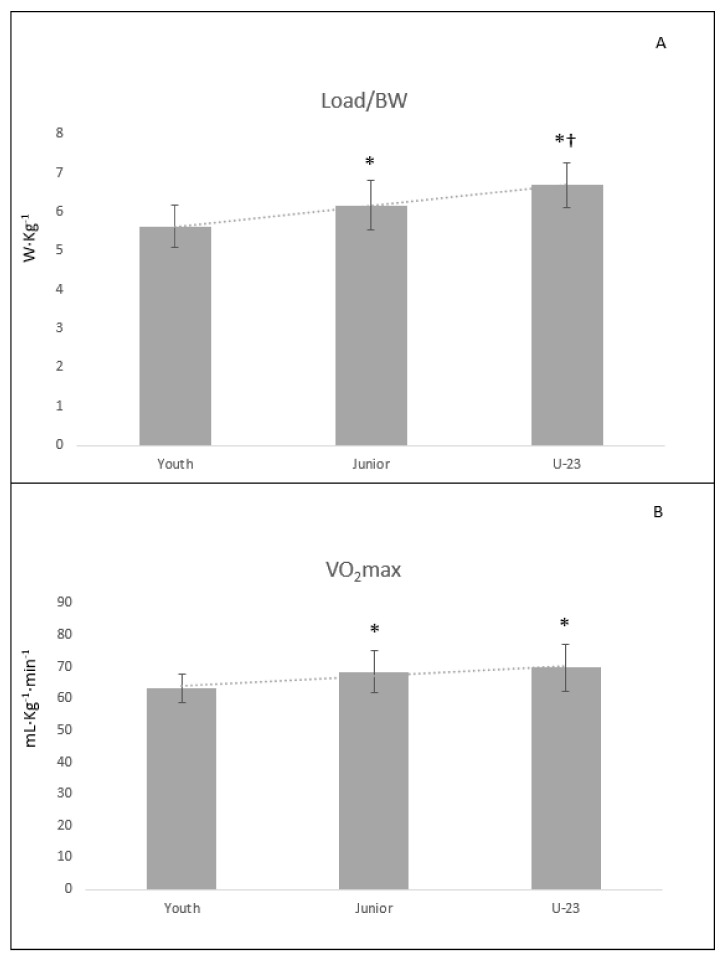
Maximal oxygen uptake (**A**) and Load/BW (**B**) values. Load/BW (**A**) = work load normalized to body weight; VO_2_max (**B**) = maximal oxygen uptake; * = Significant differences with Youth group; † = Significant differences with Junior group.

**Figure 2 biology-10-01196-f002:**
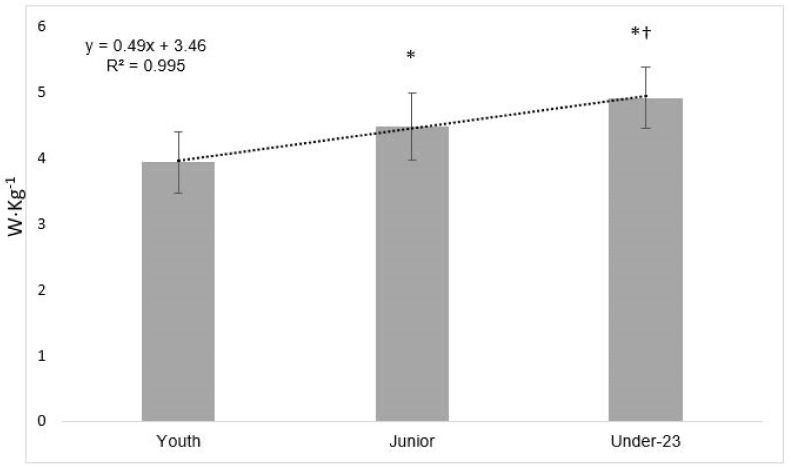
Estimated functional threshold power normalized to body weight. FTP = functional threshold power; BW = body weight; * = Significant differences with the Youth group; † = Significant differences with the Junior group.

**Table 1 biology-10-01196-t001:** General characteristics of the participants.

UCI Category	Age Range (Years)	Weight (Kg)		Height (cm)		BMI (Kg/m^2^)	
	Range	Mean	SD	Mean	SD	Mean	SD
Youth (*n* = 24)	15–16	61.2	7.4	173.2	6.4	20.4	2.3
Junior (*n* = 22)	17–18	66.5	7.4	178.2	5.9	20.9	2.3
U-23 (*n* = 15)	19–22	64.1	4.3	176.7	5.7	20.5	0.9

UCI = Union Cycliste Internationale; SD = standard deviation (±); BMI = body mass index.

**Table 2 biology-10-01196-t002:** Performance assessments data.

Zone	Variable	Youth		Junior		U-23	
		Mean	SD	Mean	SD	Mean	SD
**VT1**	**HR** (bpm)	149.5	12.2	149.7	13.2	139.7 *^,†^	11.8
	**HRmax** (%)	76.1	5.6	74.9	5.9	72.6	5.4
	**VO_2_** (L·min^−1^)	2.1	0.3	2.4 *	0.3	2.3	0.2
	**VO_2_R** (mL·Kg^−1^·min^−1^)	35.4	3.2	37.0	5.1	35.7	4.1
	**%VO_2_max** (%)	56.8	4.3	54.9	5.4	52.2 *	5.2
	**Load** (W)	152.7	20.8	179.2 *	22.3	181.3 *	22.0
	**Load/BW** (W·Kg^−1^)	2.51	0.26	2.72 *	0.40	2.84 *	0.37
**VT2**	**HR** (bpm)	183.8	10.2	185.1	9.1	179.2 ^†^	6.1
	**HRmax** (%)	93.5	2.6	93.0	2.9	93.2	2.0
	**VO_2_** (L·min^−1^)	3.3	0.4	3.9 *	0.4	3.8 *	0.4
	**VO_2_R** (mL·Kg^−1^·min^−1^)	54.2	4.3	58.6 *	7.7	59.3 *	6.9
	**%VO_2_max** (%)	87.1	5.0	86.6	4.6	86.7	5.6
	**Load** (W)	252.8	35.2	303.4 *	30.3	317.9 *	36.0
	**Load/BW** (W·Kg^−1^)	4.16	0.55	4.60 *	0.57	4.98 *	0.61

SD = standard deviation (±); VT1 = ventilatory threshold 1; VT2 = ventilatory threshold 2; HR = heart rate; VO_2_ = oxygen uptake; VO_2_R = oxygen uptake normalized to body weight; VO_2_max = maximal oxygen uptake; BW = body weight; * = Significant differences with the Youth group; ^†^ = Significant differences with the Junior group.

**Table 3 biology-10-01196-t003:** Maximal values in the VO_2_max test.

Variable	Youth		Junior		U-23	
	Mean	SD	Mean	SD	Mean	SD
**HR** (bpm)	196.5	8.8	199.0	6.6	192.5 ^†^	8.5
**VO_2_** (L·min^−1^)	3.8	0.5	4.4 *	0.3	4.4 *	0.5
**VO_2_R** (mL·Kg^−1^·min^−1^)	63.3	4.5	68.5 *	6.5	69.7 *	7.5
**RER**	1.14	0.04	1.17	0.05	1.15	0.04
**Load** (W)	343.2	45.6	407.5 *	37.6	428.3 *	37.7
**Load/BW** (W·Kg^−1^)	5.63	0.55	6.17 *	0.64	6.70 *^,†^	0.57
**Time to exhaustion** (second)	1107.2	124.7	1311.8 *	115.4	1323.7 *	125.5
**Lactate** (mmol·L^−1^)	12.6	2.8	15.6 *	4.0	14.6	2.8

SD = standard deviation (±); MAX = maximal value; HR = heart rate; VO_2_ = oxygen uptake; VO_2_R = oxygen uptake normalized to body weight; VO_2_max = maximal oxygen uptake; BW = body weight; RER = respiratory exchange ratio; * = Significant differences with Youth group; ^†^ = Significant differences with Junior group.

**Table 4 biology-10-01196-t004:** Values in the maximal fat oxidation zone.

Variable	Youth		Junior		U-23	
	Mean	SD	Mean	SD	Mean	SD
**HR** (bpm)	137.9	16.1	131.8	19.2	136.6	11.0
**HRmax** (%)	70.4	6.6	66.6	9.5	70.9	4.3
**VO_2_** (L·min^−1^)	1.9	0.3	2.1	0.4	2.2 *	0.4
**VO_2_R** (mL·Kg^−1^·min^−1^)	31.7	4.6	32.2	5.8	35.1	5.4
**%VO_2_max** (%)	51.2	9.0	48.9	8.5	51.2	5.4
**Load** (W)	147.7	30.1	163.9	35.1	181.6 *	37.6
**Load/BW** (W·Kg^−1^)	2.47	0.55	2.51	0.51	2.84	0.58
**RER**	0.88	0.03	0.87	0.03	0.89	0.03
**MFO** (g·h^−1^)	19.7	6.7	23.3	9.3	22.6	7.9

SD = standard deviation (±); HR = heart rate; VO_2_ = oxygen uptake; VO_2_R = oxygen uptake normalized to body weight; VO_2_max = maximal oxygen uptake; BW = body weight; RER = respiratory exchange ratio; MFO = maximal fat oxidation; * = Significant differences with the Youth group.

**Table 5 biology-10-01196-t005:** Estimated functional threshold power.

Variable	Youth		Junior		U-23	
	Mean	SD	Mean	SD	Mean	SD
**FTP** (W)	240.4	39.5	296.0 *	32.5	314.0 *	32.6
**FTP** (% Pmax)	69.7	2.4	72.5 *	1.3	73.2 *	1.2
**FTP/BW** (W·Kg^−1^)	3.93	0.46	4.48 *	0.51	4.91 *^,†^	0.47

SD = standard deviation (±); FTP = functional threshold power; BW = body weight; Pmax = Maximal power output; * = Significant differences with Youth group; ^†^ = Significant differences with the Junior group.

## Data Availability

The original data report is available to reviewers by contacting the corresponding author.

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
