# Peer review of "Performance Profile among Age Categories in Young Cyclists"

_biology, 2021, doi:10.3390/biology10111196_

Round 1

Reviewer 1 Report

The article “Performance Profile among Age Categories in Young Cyclists” by Marín-Pagán et al. investigates differences in the power profile of athletes in Youth, Junior and U-23 categories.

The main results are that there were apparent differences between categories in maximal values of VO2max, Pmax and functional Threshold Power and FTP/BW, with U-23 (and juniors to a lesser extent) showing higher values than Youth athletes. I think the study is very interesting and I am pleased to see some research in this area. This is a comparative study and I think it fills a gap in the comparison between young, juniors and U-23 cycling categories. The next step will certainly be a longitudinal study. I have some concerns with the statistical analyses, given the number of tests performed. I think this should be an important but easy fix and it may not change the general trends dramatically. More details on how these were done would be helpful. Another figure on VO2max and Load/BW would do a better job than a table at showing the trends through the categories.

Questions/Comments:

  • What type of body changes might affect those differences observed between categories? Muscle mass, hormonal levels? More information on this would be very interesting.
  • How many years have each participant been competing or training within a category? I expect that there is more variation in competition experience (and/or competition success) within the Youth category compared to the U-23 category. Therefore, the physiological profiles of U-23 are better indicators of “performance requirements” than the physiological profiles of Youth because U-23 have most likely been more successful in the past.
  • Can mental aspects influence the outcome of the incremental test and differ between categories?
  • Line 157 – “repeated independent t-test”. I am confused about what test the authors have used. Is it multiple independent t-tests? Or multiple repeated-measures t-tests? Those are two different tests. Repeated-measures t-tests assume that the same athletes were tested multiple times. I imagine that the authors meant the first one: multiple independent t-tests. In that case I still have some concerns with regards to multiple comparison testing. It seems that they have done 3 different tests (between each pair of categories) for 34 different variables. This is equivalent to 102 tests. With a 0.05 significant threshold, this means that 5 tests of the 102 are expected to be false positives (type I error). I strongly suggest that the authors try to get around that problem by using a Tukey's test for each dependent variable tested. Tukey's test is a t-test that compares means between category pairs and corrects for these family-wise error rates that are typical when doing multiple tests like this. This will reduce the number of tests by 3 (so down to 34). Then, I also suggest the authors to determine how many of those tests are truly significant at a False Discovery Rate (FDR) of 5% or maybe 10%. This will indicate how many were actually false positive. FDR is less stringent than Bonferroni corrections and would be appropriate I think here.
  • Line 158 – The authors seem to take the U-23 as a “reference”. Is this during the t-test? Or is it a reference for effect sizes only? I do not see any significant symbol for the Youth category so I wonder if that category was instead taken as the reference? More details on this would be very useful.
  • Line 230 – “Similar differences were found for all variables analyzed in the maximal effort zone.” Are authors referring to Table 4? Because RER was not significantly different and HR and Lactate were only significant for one of the comparisons.

Typos:

Line 21 – “to evaluate…”

Line 44 – “incremental test…” ?

Line 72 – “between cyclists, De Pauw et al. …”

Line 97 – “with heavy strength training being recommended to achieve…”

Line 120 – “University Ethical Committee…”

Line 124 – “February…”

Line 127 – “consisted of a combined test…”

Line 243 – “were found for Load/BW…”

Line 274 – “had not be reported for either differences in economy…”

Line 284 – “Additionally, FTP/BW showed…”

Line 286 – “the cycling level for cyclists in young ages categories…”? Did the authors mean “the cycling potential for cyclists…”?

Author Response

Thank you very much for your comments. Find attached our reply.

Reviewer 2 Report

The present study examined and compared the physiological profiles of different road cyclists age categories – youth, juniors and under-23. In my opinion, the manuscript was well written. Unfortunately, the manuscript required substantial English editing and spelling checking. Here, I have some suggestions for authors.

  1. The introduction needs a more vigorous justification for this study. The introduction lacks justification of the study, such as why authors want to examine the youth, juniors, and under-23 need to be stronger.  
  2. Would you mind adding a conceptual framework? 
  3. Please include sample size calculation.
  4. Would you mind stating the ethical committee code?
  5. The physiological parameters/assessments should be separated into different sections. 
  6. What is the procedure of the study? How was the data being collected? Which parameter was collected first? In what environment? What protocol was used?
  7. Please provide the data collection date. 
  8. Please check your SPSS version. SPSS 21.0 is no more valid for a long time.
  9. Why use a repeated independent t-test? 
  10. Please analyse and report the age, weight, height and BMI between the three groups and report the p-value with effect size. 
  11. Please report the actual p-value and effect size for the results in Table 2, Table 3, and Table 4.  
  12. Discussion should be discussed comparing with the previous study and not mostly about the results. Please compare with previous literature.
  13. Please add limitations of the study.
  14. Overall, I felt the study needed stronger justification, conceptual framework etc. Thank you. 

Author Response

(The authors gave the same response as above.)

Reviewer 3 Report

Dear Authors,

You have written an interesting study. The introduction is clearly written with up to date literature.

Methods section:

Participants: please report the training experience of athletes

Assessments: How were height and weight measured? Report

At what part of the day were the tests performed? report

What were the instructions to participants 1 day before the test? Report

The discussion is not well develped and the limitations paragraph is missing. Amend accordingly

Overall the paper is clearly written, however the discussion does not really explain possible diferences between age groups.

To summarise: he paper looks promising, therefore I recomend major revision.

Kind regards

Author Response

(The authors gave the same response as above.)

Round 2

Reviewer 1 Report

The legend for the new Figure 1 is reversed for panel A and B, should be: "Load/BW (A) and Maximal Oxygen Uptake (B) values".

Thank you for taking my comments into consideration and correcting the "repeated independent t-test". As for the tests, the authors say that "..., in the discussion of the results we relied more on the magnitude of the changes (i.e., Effect sizes) than the p-value itself". I recommend the authors to explain this in their discussion or emphasize more on the effect sizes that are now included in the manuscript.

Author Response

Thank you very much for your comments. The manuscript has been better after the review process. We have added the proposed changes to the manuscript.

Reviewer 2 Report

I had checked and the authors had completed all the comments as suggested and thus I propose to accept this manuscript. 

Author Response

Thank you very much for your comments. The manuscript has been better after the review process.

Reviewer 3 Report

Dear Authors

Thank you for addressing all of my questions and recommendations. From my point of view, the paper is prepared for acceptance. Therefore, I recommend to the editor acceptance in present form.

Kind regards

Author Response

(The authors gave the same response as above.)
